# Elevated Glycated Haemoglobin (HbA1c) Is Associated with an Increased Risk of Pancreatic Ductal Adenocarcinoma: A UK Biobank Cohort Study

**DOI:** 10.3390/cancers15164078

**Published:** 2023-08-13

**Authors:** Declan McDonnell, Adrian W. E. Cheang, Sam Wilding, Sarah H. Wild, Adam E. Frampton, Christopher D. Byrne, Zaed Z. Hamady

**Affiliations:** 1Human Development & Health, University of Southampton, University Hospital, Southampton SO16 6YD, UK; a.w.e.cheang@soton.ac.uk (A.W.E.C.); s.a.wilding@soton.ac.uk (S.W.); c.d.byrne@soton.ac.uk (C.D.B.); z.hamady@soton.ac.uk (Z.Z.H.); 2HPB Unit, University Hospital Southampton, Southampton SO16 6YD, UK; 3Usher Institute, University of Edinburgh, Edinburgh EH8 9YL, UK; sarah.wild@ed.ac.uk; 4Section of Oncology, University of Surrey, Guildford GU2 7XH, UK; adam.frampton@surrey.ac.uk; 5HPB Unit, Royal Surrey NHS Foundation Trust, Guildford GU2 7XX, UK

**Keywords:** pancreatic, adenocarcinoma, PDAC, diabetes, NODM, HbA1c, T3cDM

## Abstract

**Simple Summary:**

Pancreatic cancer is associated with a poor prognosis. This is often because it is diagnosed when it is too late for potentially curative treatment. There is an established link between raised blood sugars and pancreatic cancer. HbA1c is a blood test which provides clinicians with an average measurement of blood sugar over the past month. It is unclear what is the strength of the association between elevated HbA1c using the same parameters to diagnose prediabetes and diabetes, and if this association changes with length of follow up time. This study demonstrates that a HbA1c value consistent with a new diagnosis of diabetes (≥48 mmol/mol) is associated with a greater than eight fold risk of being diagnosed with pancreatic cancer in the next 12 months compared to a HbA1c in the normal range (<42 mmol/mol).

**Abstract:**

Background: The role of dysglycaemia as a risk marker for Pancreatic Ductal Adenocarcinoma (PDAC) is uncertain. We investigated the relationship between glycated haemoglobin (HbA1c) and incident PDAC using a retrospective cohort study within the UK Biobank. Methods: A study involving 499,804 participants from the UK Biobank study was undertaken. Participants were stratified by diabetes mellitus (DM) status, and then by HbA1c values < 42 mmol/mol, 42–47 mmol/mol, or ≥48 mmol/mol. Cox proportional hazard models were used to describe the association between HbA1c category (with time-varying interactions) and incident PDAC. Results: PDAC occurred in 1157 participants during 11.6 (10.9–12.3) years follow up [(median (interquartile range)]. In subjects without known DM at baseline, 12 months after recruitment, the adjusted hazard ratios (aHR, 95% CI) for incident PDAC for HbA1c 42–47 mmol/mol compared to HbA1c < 42 mmol/mol (reference group) was 2.10 (1.31–3.37, *p* = 0.002); and was 8.55 (4.58–15.99, *p* < 0.001) for HbA1c ≥ 48 mmol/mol. The association between baseline HbA1c and incident PDAC attenuated with increasing duration of time of follow-up to PDAC diagnosis. Conclusions: Dysglycaemia detected by elevated HbA1c is associated with an increased risk of PDAC. The strength of the association between elevated HbA1c and incident PDAC is inversely proportional to the time from detecting dysglycaemia but remains significant for at least 60 months following HbA1c testing.

## 1. Introduction

Pancreatic cancer is the fourth most common cause of cancer-associated deaths in the UK [1]. Over 90% of pancreatic cancers are pancreatic ductal adenocarcinomas (PDAC) [2]. Pancreatic neuroendocrine neoplasms (PanNEN) are the next most common subtype [3]. They are rare and more indolent than PDAC, may present in a different manner, with better long term survival, and therefore are not considered in this study [4]. The terms “pancreatic cancer” and “PDAC” are often used interchangeably. This will also be the case for this study. PDAC is associated with very poor survival and current overall one and three year survival is 22.5% and 6.6%, respectively. Importantly, overall survival is higher (52.7% and 27.1% at one and three years, respectively) when the cancer is identified and treated early, before the disease spreads [5]. This finding emphasises the importance of early detection, and therefore more effective treatment for PDAC.

Diabetes mellitus (DM) has a well-established link to PDAC [6,7], and the complex, bidirectional relationship between the two conditions is known as reverse causation or “dual causality” [8]. Impaired plasma glucose regulation, or dysglycaemia, has been demonstrated in approximately half of those diagnosed with PDAC. In one study, 17/30 (57%) of those with DM of less than two years duration had improvement in their plasma glucose regulation following resection of the cancer [9]. This suggests a potential paracrine effect of the PDAC, whereby potential mediators of DM, such as adrenomedullin, are locally produced by the tumour and inhibit the action of β-cells in the pancreas [10]. Overall, DM is associated with 1.5–2 fold higher risk of PDAC, compared to people without DM [11,12]. Furthermore, DM diagnosed within the 24–36 months prior to PDAC diagnosis (new onset diabetes mellitus/NODM) has a particularly strong association with incident PDAC [13,14,15,16]. NODM may be a useful component of screening for PDAC. In one study, 18 (0.85%) people with NODM identified between 1950 and 1994 in a population-based cohort of 2122 Rochester, Minnesota residents aged ≥ 50 years over developed PDAC within three years of a DM diagnosis [17]. The results were compared to the state registry data and the observed-to-expected ratio of PDAC in the cohort was 7.94 (95% CI: 4.70–12.55).

Glycated haemoglobin (HbA1c) has excellent clinical utility for the diagnosis of dysglycaemia, with values ≥ 48 mmol/mol diagnostic for DM [18]. Raised HbA1c is associated with an increased risk of PDAC, with a 1 mmol/mol increase in HbA1c in the previous year to have an adjusted odds ratio (OR) for PDAC of 1.06 (95% CI: 1.06–1.07) [19]. Tan et al. undertook a population-based nested case-control study using the QResearch primary care database (V.45) of 28,137 PDAC and 261,219 matched controls in England [20]. They were able to demonstrate an increased risk of PDAC with NODM (<3 years duration) having an OR 4.93 (95% CI: 4.69–5.18), compared to long-standing DM with an OR 1.89 (95% CI: 1.82–1.97). In addition to this, an increase in HbA1c within the 33 months prior to PDAC diagnosis was also associated with PDAC, particularly in those with established DM; although the magnitude of the increase in risk with HbA1c was not quantified. This study also established prediabetes diagnosed within the previous three years to be associated with PDAC given the OR 1.63 (95% CI: 1.50–1.77), although it is not clear how this diagnosis of prediabetes was established. Subsequently, the relationship between nondiabetic hyperglycaemia (i.e., HbA1c between 42–47 mmol/mol, or prediabetes) with a risk of incident PDAC remains uncertain. Understanding the role of these intermediate values of HbA1c is particularly important given the wider use of HbA1c measurement to assess levels of glycaemia. For example, in England, people aged 40–74 years have access to HbA1c measurements as part of the NHS health check and diabetes prevention program (DPP) [21].

The UK Biobank (UKBB) is a large prospective cohort study of over 500,000 participants which aims to identify and refine novel and known risk factors for a range of conditions [22]. A comprehensive description of the design of the UKBB was published in 2014 [23]. The UKBB cohort has been used to highlight the increased risk of breast cancer with DM [24], as well as the increased risk of colorectal cancer with obesity [25]. UKBB is updated periodically using a range of resources such as cancer registries to identify incident disease conditions. There is already an established association between both DM [26] and elevated HbA1c [27] and subsequent PDAC within the UKBB population. Ke et al. [26] noted a history of DM among 728 incident PDAC patients more than doubled the risk of incident PDAC when compared to a control group without neoplasms or nonmalignant neoplasms with a risk ratio (RR, 95% CI) of 2.08 (1.64–2.63). Rentsch et al. [27] looked at the risk of incident PDAC based on HbA1c over a median 7.1 years (IQR 6.4–7.7 years) follow-up after adjusting for age, sex (except sex-specific cancers), ethnicity, deprivation, BMI, physical activity, cardiovascular and DM diagnoses at baseline, smoking status and alcohol consumption. They compared HbA1c ≥ 55 mmol/mol with the UKBB cohort median HbA1c of <35 mmol/mol and established a HR (95% CI) of 1.55 (1.22–1.98). However, it remains unclear whether there is an association between conventional HbA1c categories and incident PDAC among people without previously diagnosed DM. It is also uncertain if the strength of the association between NODM and subsequent PDAC is independent of obesity, which is associated with both PDAC [28] and DM [29,30]. Unintentional weight loss is also associated with both conditions (when glucose levels are very high prior to a diagnosis of DM) [31].

The primary aim of this UKBB retrospective cohort study was to investigate the time-varying association between categories of HbA1c at baseline with incident PDAC in those with and without DM at enrolment to UKBB. The study is described using the STROBE cohort reporting guidelines.

## 2. Materials and Methods

### 2.1. Identification of Incident PDAC Participants and the Main Exposures

Between 2006 and 2010, UKBB recruited 502,413 participants aged between 37 and 73 years of age across 22 study centres in the UK. Data on cancer diagnoses for participants resident in England and Wales is provided to UKBB by the Medical Research Information Service, based at the National Health Service Information Centre (http://www.ic.nhs.uk/services/medical-research-informationservice). The Information Services Division (http://www.isdscotland.org/HealthTopics/Cancer/) and its successor Public Health Scotland provides UKBB with the cancer data records for participants resident in Scotland. People with a history of pancreatic cancer were identified at cohort enrolment, and incident cancers were identified throughout the follow up period up to 25 September 2020, the most recent date for which cancer incidence data are complete. All pancreatic cancers (encompassing PDACs and PanNENs) were identified using the International Classification of Diseases, tenth revision (ICD-10) using the codes C250-259. Participants were excluded from the study if they had not indicated whether or not they had doctor-diagnosed DM or were taking medication for DM (n = 2609); if they had a previous diagnosis of pancreatic cancer (n = 55) at baseline; or if the final histology of an incident pancreatic tumour confirmed a PanNEN (ICD-10 C254 and/or an ICD-0-3 histology code corresponding to PanNEN) (n = 73). By the end of follow-up, there were 1157 participants with incident PDAC within UKBB identified during follow up (Figure 1). Participants contributed person–time data from the date of attending the UKBB Assessment Centre for their baseline assessment until the earliest of: date of PDAC diagnosis, death (leading to withdrawal from the study) or end of follow up (25 September 2020).

The primary outcome was incident PDAC. History of DM at cohort enrolment was defined by participants reporting that a doctor told them they had DM or that they were prescribed pharmacological treatment for DM. HbA1c was available as a continuous variable and was categorised into one of three ranges: HbA1c < 42 mmol/mol, HbA1c 42–47 mmol/mol and HbA1c ≥ 48 mmol/mol, according to established thresholds for defining normoglycaemia, prediabetes (non-diabetic hyperglycaemia) and type two DM in people not previously known to have DM [18].

Ethical approval for the UKBB was obtained from the North West Multi-Centre Research Ethics Committee (reference number 06/MRE08/65), the National Information Governance Board for Health and Social Care in England and Wales and the Community Health Index Advisory Group in Scotland. When recruited, participants gave informed, written consent for participation and follow-up. Participants were able to withdraw their consent for follow-up, subsequently. Baseline information was gathered from the participants using a self-assessment questionnaire (available online: http://www.ukbiobank.ac.uk/wp content/uploads/2011/06/Touch_screen_questionnaire.pdf?phpMyAdmin=trmKQlYdjjnQIgJ%2CfAzikMhEnx6) which enquired about sociodemographics, medical history, lifestyle exposures and medication use. Anthropometric investigations and baseline blood tests were taken on attendance to the regional assessment centre before the latter were sent for processing and storage at the central laboratory [32].

Body weight and bioelectrical impedance analysis (BIA) were measured using the Tanita BC-418MA body composition analyser. Height was recorded using a Saca 202 device and BMI calculated by dividing the weight in kilograms by the square of the height in metres [25].

### 2.2. Statistical Analysis

Normally distributed continuous data are summarised as means and standard deviations (SDs). Categorical data are summarised using proportions, comparing those with and without incident PDAC at the end of follow-up. The risk of developing PDAC was determined for each variable using hazard ratios (HR) and 95% confidence intervals (CIs) estimated using Cox Proportional Hazards Models. The reference category for HbA1c was <42 mmol/mol. The proportional hazards assumption was tested using Schoenfeld residuals and there was no evidence of nonproportionality detected for HbA1c in those with DM on enrolment. However, model assumptions were violated for HbA1c in those without history of DM. Consequently, time-varying coefficient interaction terms were created between HbA1c and exposure time (one for 42–47 mmol/mol and time between enrolment and censor date, and one for ≥48 mmol/mol and time between enrolment and censor date) amongst participants who did not have a confirmed diagnosis of DM on enrolment. This approach was undertaken to determine the risk of PDAC according to different levels of baseline HbA1c and increasing duration of follow up. The time between enrolment and censor date was measured over one year time increments.

Identifying modifiable and non-modifiable risk factors for PDAC within UKBB emphasises the importance of addressing those with the potential to reduce the risk of PDAC if adjusted accordingly [26]. Modifiable covariables and potential confounders that were measured at baseline were selected for inclusion in the models based on their established association with PDAC, and included: obesity (measured by Body Mass Index (BMI) [28]), weight change in the previous 12 months [33], tobacco smoking (never, ex, current and <20/day, and current and ≥20/day) [34,35], alcohol intake (never, special occasions, one to three times/month, one to two times/week, three to four times/week, daily) [36,37] and processed meat consumption (never, less than once a week, once a week, two to four times per week, and five or more times per week) [38]. The nonmodifiable variables ethnic background (White, Mixed, Asian or Asian British, Black or Black British, Chinese, Other ethnic group) [39], age and sex were also included in models.

BMI was categorised using ethnic-specific thresholds to define underweight, normal (reference category), overweight and obese groups [40]. The other reference categories for the covariables were: nonsmoker, not consuming alcohol, not consuming processed meat, no change in weight compared to the previous 12 months, and ethnic background “White”, as 95% of all UKBB participants identified this way. All statistical analyses were conducted using Stata v.16 (College Station, TX, USA, StataCorp LLC).

## 3. Results

Basic participant characteristics are described in Table 1, with an extended version available in Appendix A. There were 473,264 people without DM (94.7%), and 26,540 with DM (5.3%) on enrolment to UKBB. Some 2609 (0.5%) of participants did not know or preferred not to give their DM status at enrolment and were excluded. There were 1157 participants who developed incident PDAC during the median time from enrolment to censor date of 11.6 (10.9–12.3) years. One thousand and thirty-one of these participants with incident PDAC did not have DM on enrolment compared to one hundred and twenty-six who did have DM. Mean (SD) age at baseline was 56.4 (SD 8.1) years for those without DM on enrolment and 59.5 (SD 7.2) years for those with a history of DM. Men constituted 44.7% of those without DM on enrolment, and 60.6% of those with a history of DM. People of white ethnicity composed 94% of the cohort. HbA1c was ≥48 mmol/mol in 3360 (0.8%) of those without established DM on enrolment, and in 14,038 (57.5%) of those with DM on enrolment (*p* < 0.001). BMI was higher in people with DM on enrolment at 31.3 (SD 5.9) kg/m^2^ compared to those without DM at 27.2 (SD 4.6) kg/m^2^ (*p* < 0.001).

Univariable analyses by DM status at enrolment are described in Appendix A. Among people without DM at enrolment, there was an association between HbA1c category and incident PDAC, but the proportional hazards assumption was violated. This violation suggests that the association between elevated HbA1c and PDAC was non-linear (i.e., a significant number of those with dysglycaemia, detected by elevated HbA1c, would reach the censor date, i.e., being diagnosed with PDAC early in the follow up period). Obesity, former and current smoking, processed meat consumption, increasing age and male sex were also associated with a higher risk of incident PDAC in people without DM at enrolment. Those participants from an Asian or Asian British background had a reduced association with PDAC. Weight gain appeared to be protective against incident PDAC but hazards were also not proportional. In people known to have DM at enrolment, HbA1c ≥ 48 mmol/mol, current smoking, processed meat consumption and increasing age were associated with a higher risk of PDAC, and none of the proportional hazards assumptions were violated.

Adjustment for BMI category, weight change compared to one year prior to enrolment, smoking status, alcohol consumption, processed meat intake, age and sex in multivariable models attenuated the hazard ratios for HbA1c category for incident PDAC among people without DM at enrolment but had little effect on the estimates among people known to have DM (Table 2). The proportional hazards assumption remained violated for people not known to have DM at enrolment. Therefore, a time-varying coefficient interaction for this group stratified by 12-monthly time intervals was created. Table 3 shows the association between HbA1c category 42–47 mmol/mol and incident PDAC after multivariable adjustment. Table 4 shows the association between HbA1c category ≥ 48 mmol/mol and incident PDAC after multivariable adjustment. Figure 2 shows these results plotted on a scatter chart for comparison. The Stata syntax for this time-varying coefficient is available in Appendix A. There was a particularly strong association with risk of incident PDAC within the first 12 months from enrolment for both HbA1c 42–47 mmol/mol at 2.10 (1.31–3.37, *p* = 0.002) and HbA1c ≥ 48 mmol/mol at 8.55 (4.58–15.99, *p* < 0.001), respectively). HRs attenuated over the first 72 months from enrolment for the HbA1c 42–47 mmol/mol group and over the first 60 months from enrolment for the ≥48 mmol/mol group. This supports an association between NODM and PDAC.

## 4. Discussion

Our original data show an association between HbA1c ≥ 48 mmol/mol and a higher risk of incident PDAC in UKBB participants regardless of known DM status at cohort enrolment. There was also an association between HbA1c 42–47 mmol/mol and incident PDAC in those without a previous DM diagnosis. The association between either category of dysglycaemia and incident PDAC among people not previously known to have DM was particularly strong for PDAC diagnosed within 12 months of cohort enrolment, and the strength of the association declined with increasing follow-ups. These findings highlight the potential for a bidirectional relationship between dysglycaemia and PDAC with potential reverse causality in the early years of follow-up. The association between HbA1c and incident PDAC was independent of BMI and other known risk factors for PDAC. In those who had DM on enrolment, the hazard associated with a higher HbA1c was approximately constant during follow-up and did not violate the proportional hazards assumption. This confirms the established risk between DM and PDAC. In contrast, in those without DM on enrolment, the proportional hazards assumption was violated with an inverse association between hazard ratios and duration of follow-up. This necessitated the use of a time-varying coefficient (see Appendix A), which accounted for the increased risk of incident PDAC in those with dysglycaemia diagnosed by a raised HbA1c on enrolment to UKBB. This novel methodological approach has not been used for the UKBB dataset before and provides a strategy that might be applied to UKBB and similar datasets when the proportional hazards assumption is violated. This is particularly important when the strength of the association attenuates with time, such as when an exposure and outcome may be affected by reverse or dual causality.

An association between nondiabetic hyperglycaemia and PDAC has previously been described in the European Prospective Investigation into Cancer and Nutrition (EPIC) study [41]. This nested case-control study of 519,978 men and women of 35–70 years of age used the Diabetes Control and Complications Trial (DCCT) classification for HbA1c to categorise glycaemia. In this study, ORs (95% CI) for HbA1c (6.0–6.4%/42–46 mmol/mol) were compared to group with a lower reference range than our study (4.8–5.4%/29–36 mmol/mol), and incident PDAC was 1.65 (1.01–2.70). Additionally, fasting blood glucose (FBG) above the cut-point for diagnosis of DM (126 mg/dL/7.0 mmol/L) is also known to be associated with an increased risk of incident PDAC [15]. An association between impaired FBG (≥6.1–7.0 mmol/L) and an increased risk for PDAC of 1.77 (1.05–2.99) compared to those with an FBG < 6.1 mmol/L has been described [42]. Measuring dysglycaemia using HbA1c does not require a fasting blood sample and so has much greater utility in routine clinical practice than measuring FBG. Our study is the first to demonstrate this association between nondiabetic hyperglycaemia, clearly defined as HbA1c 42–47 mmol/mol in persons without a history of DM, with incident PDAC. This builds upon the work by Tan et al. [20] but provides a clearer definition of prediabetes.

The attenuation of strength of the association between DM and incident PDAC over time has been noted in other studies. Ben et al. undertook a case-control study of the blood glucose levels from 1458 (48.8%) people with a diagnosis of PDAC and 1528 age- and sex-matched controls identified from one of two university-affiliated hospitals in Shanghai, China. The authors reported adjusted odds ratios for PDAC (95% CI) of 4.43 (3.44–5.72) in those with DM of less than two years since diagnosis and of 2.11 (1.51–2.94) for those with DM duration of more than two years compared to controls without DM [43]. Danker et al. analysed time varying associations with glycaemic status in an Israeli population-based sample of 4097 pancreatic cancer patients among 2,186,196 men and women adjusting for age, ethnic origin, and socioeconomic status. They described the risk of pancreatic cancer to be approximately 14 and 15 times greater in men and women, respectively, in the first year following a DM diagnosis which decreased to 3.5–5.4 fold for the following year and then remained around 3 fold for the remainder of the 11 year follow up period compared to people without DM [44]. Our study is consistent with these findings in that the greatest risk of PDAC was within the first 12 months following a DM diagnosis, which also attenuated over time, but the study by Danker et al. demonstrated a higher relative risk over a longer term of follow-up. It is difficult to make direct comparisons between the two studies as ours was focused on the risk of PDAC following establishment of dysglycaemia from HbA1c, whereas the focus in the study by Danker et al. was to investigate the risk of pancreatic cancer following a confirmed DM diagnosis. Danker et al. used a variety of methods to diagnose DM, including HbA1c, as well as two FBG measurements >125 mg/dL within a 12-month period or a 2 h plasma glucose level >200 mg/dL during a formal oral glucose tolerance test. It is possible that the approach to diagnosing DM may account for the discrepancies between the study findings. Our study is the first to note an annual attenuation of the association between HbA1c measured in mmol/mol and incident PDAC in those without previously established DM. It is also unique in demonstrating the attenuation over time in those with nondiabetic hyperglycaemia and incident PDAC.

The limited availability of repeat exposure measurements between recruitment and development of an outcome is a major limitation within the UKBB study. Although UKBB regularly collates data from hospital episode statistics and cancer registries to record incidence of medical conditions, only a small proportion of participants have repeat measurements of the exposures of interest. Therefore, we were unable to use time-updated variables in the regression analysis to study the risk of incident PDAC with a change in the exposure measurements during follow-up. If repeated exposure measurements were available, the proportional hazards assumption may not have been violated for those who were found to have dysglycaemia without a history of DM. Other prospective studies such as the UK Early Detection Initiative (UK-EDI) and Enriching New-Onset Diabetes for Pancreatic Cancer (ENDPAC) undertake regular measurements of their whole cohorts to determine how variations in factors such as blood glucose regulation, weight changes and age impact on the risk of incidental PDAC [45,46].

Approximately 95% of the UKBB population come from a “White” ethnic background. This is a major limitation, as the UK census performed close to the time UKBB finished recruitment shows 86% of the residents of England and Wales identified as White [47]. Other population studies that have been performed in more heterogenous groups show that non Hispanic White participants (making up 45.1% of their study) are at the highest risk at 2.37 (95% CI, 1.75–3.14) per 1000 person years. The relatively low number of Asian or British Asian participants within UKBB (2.0%) is not representative of the UK population at that time (8% according to the 2011 census). Subsequently, the protective effect seen in participants from this ethnic background should consider that. Additionally, this protective effect was not present after multivariable adjustment (Appendix A).

Another limitation of this study is that the blood tests taken at enrolment were random rather than fasted samples; therefore, we were unable to assess the association between FBG and PDAC within this population. It is important to consider there is an element of selection bias within UKBB and one of the major limitations is that participants tend to be healthier and more motivated to take control of their health by entering into studies like this than the population they represent [48]. Nevertheless, as this study population ages, we would also anticipate a greater incidence of both DM and PDAC to be established. Further studies on this population could explore the temporal association between confirmed DM and PDAC diagnoses. The association between PDAC and DM within this study is similar to that seen in Ke et al. [26], although their study compared risk ratios for PDAC in those with DM on enrolment compared to those without DM, after adjusting for age and sex.

The relationship between HbA1c levels and overall survival predictions would be an interesting association to explore within UKBB. However, some of the data that would be essential to study this in detail such as PDAC stage and which treatments had been utilized (surgical intervention, chemotherapy) have not yet been released by UKBB. We recognise this is a limitation of this study and hope to conduct this analysis in the future when the data are available. Additionally, we hope to apply the methods developed during this study, such as using a time-varying coefficient to a large independent patient dataset to validate the findings.

## 5. Conclusions

Our study has identified that dysglycaemia determined by HbA1c concentrations is associated with increased incident PDAC regardless of known DM status. There was a particularly strong association between nondiabetic hyperglycaemia (42–47 mmol/mol) and NODM (≥48 mmol/mol, in those without a prior history of DM) with incident PDAC in the first 12 months following a dysglycaemia diagnosis. The strength of the association between the HbA1c parameters and incident PDAC attenuated over time, which caused the proportional hazards assumption to be violated and necessitated the creation of a time-varying coefficient. This finding remained significant for at least 60 months following the diagnosis of dysglycaemia in those without an established history of DM. Compared to other studies, we have demonstrated the parameters of HbA1c and their associated risk of PDAC. This may transfer into clinical practice to facilitate early detection of PDAC through the identification of dysglycaemia or NODM. Further research is required to establish how changes in HbA1c and change in body weight over time could be used to identify people at risk who might benefit from screening for PDAC. However, the challenges with screening for PDAC need to take into account the relatively low incidence of this disease.

## Figures and Tables

**Figure 1 cancers-15-04078-f001:**
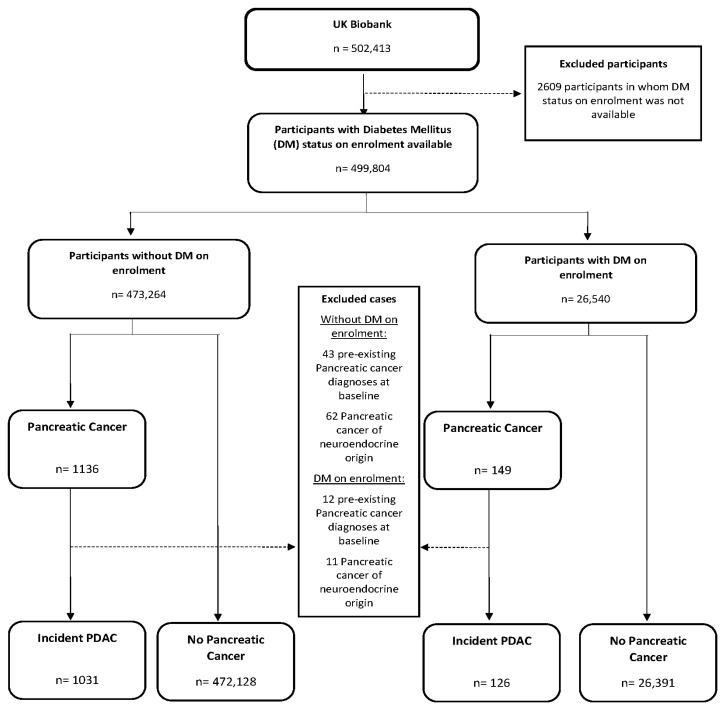
Flow diagram used to identify eligible incident pancreatic ductal adenocarcinoma cases and other participants by diabetes mellitus status within UK Biobank.

**Figure 2 cancers-15-04078-f002:**
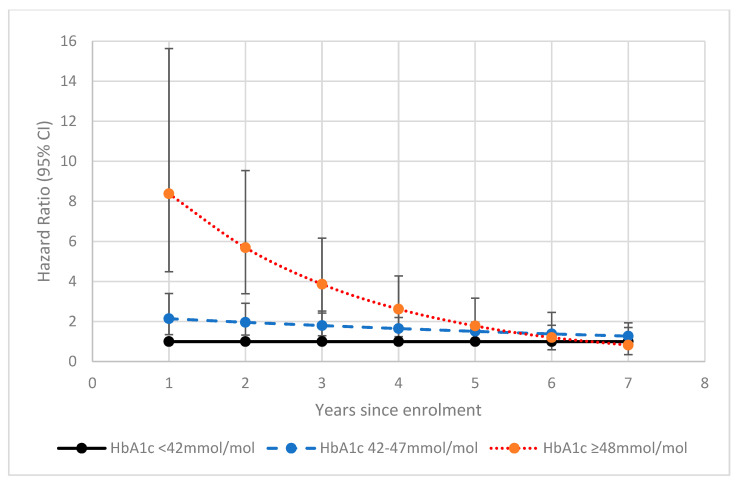
Risk of incident pancreatic ductal adenocarcinoma (PDAC) according to the time-varying coefficient interaction between glycated haemoglobin and years from enrolment into UK Biobank. Multivariable model adjusted for body mass index category, weight change compared to 1 year ago, smoking status, alcohol consumption, processed meat intake, ethnic background, age and sex. This figure highlights the significant risk of PDAC (hazard ratio 8.55, 95% CI 4.58–15.99, *p* < 0.001) when the HbA1c ≥ 48 mmol/mol in the first 12 months after enrolment.

**Table 1 cancers-15-04078-t001:** Characteristics of study participants at baseline stratified by diabetes mellitus status at enrolment to UK Biobank.

Participant Characteristics	No Diabetes Mellitus on Enrolment	Diabetes Mellitus on Enrolment	Total	*p* Value
Number of Participants, n (%)	473,264 (94.7%)	26,540 (5.3%)	499,804	<0.001 ^b^
Age at attendance (years) ^a^	56.4 (8.1)	59.5 (7.2)	56.5 (8.1)	<0.001 ^b^
Men, n (%)	211,583 (44.7%)	16,095 (60.6%)	227,678 (45.6%)	<0.001 ^c^
Glycated haemoglobin (HbA1c) category (mmol/mol), n (%)				<0.001 ^c^
<42 mmol/mol	420,780 (95.6%)	5208 (21.3%)	425,997 (91.7%)	
42–47 mmol/mol	15,852 (3.6%)	5186 (21.2%)	21,038 (4.5%)	
≥48 mmol/mol	3360 (0.8%)	14,038 (57.5%)	17,398 (3.8%)	
Body mass index (BMI) ^a^	27.2 (4.6)	31.3 (5.9)	27.4 (4.8)	<0.001 ^b^
Body mass index (BMI) category, n (%)				<0.001 ^c^
Underweight ^d^	2574 (0.6%)	36 (0.1%)	2610 (0.5%)	
Normal ^d^	156,970 (33.3%)	2838 (10.8%)	159,808 (32.1%)	
Overweight ^d^	202,425 (43.0%)	8961 (34.2%)	211,386 (42.5%)	
Obese ^d^	108,990 (23.1%)	14,381 (54.9%)	123,371 (24.8%)	

^a^ Mean average (standard deviation); ^b^ Student’s *t* test for continuous data; ^c^ χ^2^ test for categorical variables; ^d^ adjusted for variations in ethnic-specific variations. Underweight < 18.5 kg/m^2^, normal 18.5–24.9 kg/m^2^ (18.5–22.9 kg/m^2^ in Asian population), overweight = 25.0–29.9 kg/m^2^ (23–27.4 kg/m^2^ in Asian population), obese ≥ 30.0 kg/m^2^ (27.5 kg/m^2^ in Asian population).

**Table 2 cancers-15-04078-t002:** Multivariable risk of incident pancreatic ductal adenocarcinoma (PDAC) according to baseline glycaemic status in subjects with and without diabetes mellitus at enrolment to UK Biobank ^a^.

Variable	Diabetes Mellitus on Enrolment	Participants	Incident PDAC (n, %)	Incidence (per 1000 Person-Years)	Multivariable Hazard Ratio (95% CI) ^b^	*p* Value	Proportional Hazards Assumption
Glycated haemoglobin (HbA1c)							
<42 mmol/mol	No	420,789	869 (91.1%)	0.17	1 (Reference) (N/A ^c^)		<0.001
42–47 mmol/mol		15,872	66 (6.9%)	0.3	1.39 (1.07–1.81) (N/A ^c^)	0.015	
≥48 mmol/mol		3360	19 (2.0%)	0.33	2.17 (1.37–3.44) (N/A ^c^)	0.001	
<42 mmol/mol	Yes	5208	16 (13.8%)	0.28	1 (Reference)		0.2263
42–47 mmol/mol		5186	22 (19.0%)	0.39	1.28 (0.66–2.46)	0.463	
≥48 mmol/mol		14,038	78 (67.2%)	0.51	1.95 (1.12–3.37)	0.017	

^a^ Participants were asked at enrolment if they had ever been diagnosed with diabetes by a doctor or indicated they were taking medication for diabetes; ^b^ multivariable model adjusted for body mass index category, weight change compared to 1 year ago, smoking status, alcohol consumption, processed meat intake, ethnic background, age and sex. ^c^ N/A due to violation of proportional hazards assumption.

**Table 3 cancers-15-04078-t003:** Risk of incident pancreatic ductal adenocarcinoma (PDAC) according to the interaction between glycated haemoglobin (42–47 mmol/mol) and 12-monthly time intervals from enrolment in participants without established diabetes mellitus into UK Biobank ^a^.

Time Since Enrolment in UKBB	Glycaemic Category at Enrolment and Cox Proportional Hazard Ratio for Incident PDAC							
	<42 mmol/mol				42–47 mmol/mol			
	Total at Risk During Time Interval (n)	Incident PDAC during Time Interval (n)	Hazard Ratio (95% CI) ^b^	*p* Value	Total at Risk During Time Interval (n)	Incident PDAC during Time Interval (n)	Hazard Ratio (95% CI) ^b^	*p* Value
12 months	420,707	36	1 (Reference)		15,840	9	2.10 (1.31–3.37)	0.002
24 months	419,980	50	1 (Reference)		15,752	3	1.92 (1.29–2.88)	0.001
36 months	418,816	66	1 (Reference)		15,645	5	1.76 (1.26–2.49)	0.001
48 months	417,318	71	1 (Reference)		15,532	9	1.62 (1.21–2.17)	0.001
60 months	415,620	78	1 (Reference)		15,394	5	1.49 (1.14–1.94)	0.004
72 months	413,705	86	1 (Reference)		15,260	5	1.36 (1.04–1.78)	0.024
84 months	411,526	103	1 (Reference)		15,118	7	1.25 (0.93–1.68)	0.141

^a^ Participants were asked at enrolment if they had ever been diagnosed with diabetes by a doctor or indicated they were taking medication for diabetes; ^b^ multivariable model adjusted for body mass index category, weight change compared to 1 year ago, smoking status, alcohol consumption, processed meat intake, ethnic background, age and sex.

**Table 4 cancers-15-04078-t004:** Risk of incident pancreatic ductal adenocarcinoma (PDAC) according to the interaction between glycated haemoglobin (≥48 mmol/mol) and 12-monthly time intervals from enrolment in participants without established diabetes mellitus into UK Biobank ^a^.

Time Since Enrolment in UKBB	Glycaemic Category at Enrolment and Cox Proportional Hazard Ratio for Incident PDAC							
	<42 mmol/mol				≥48 mmol/mol			
	Total at Risk During Time Interval (n)	Incident PDAC during Time Interval (n)	Hazard Ratio (95% CI) ^b^	*p* Value	Total at Risk during Time Interval (n)	Incident PDAC during Time Interval (n)	Hazard Ratio (95% CI) ^b^	*p* Value
12 months	420,707	36	1 (Reference)		3356	6	8.55 (4.58–15.99)	<0.001
24 months	419,980	50	1 (Reference)		3330	2	5.80 (3.45–9.75)	<0.001
36 months	418,816	66	1 (Reference)		3312	3	3.94 (2.46–6.29)	<0.001
48 months	417,318	71	1 (Reference)		3290	3	2.67 (1.63–4.37)	<0.001
60 months	415,620	78	1 (Reference)		3265	1	1.81 (1.01–3.24)	0.045
72 months	413,705	86	1 (Reference)		3232	1	1.23 (0.60–2.50)	0.568
84 months	411,526	103	1 (Reference)		3204	1	0.82 (0.35–1.98)	0.681

^a^ Participants were asked at enrolment if they had ever been diagnosed with diabetes by a doctor or indicated they were taking medication for diabetes; ^b^ multivariable model adjusted for body mass index category, weight change compared to 1 year ago, smoking status, alcohol consumption, processed meat intake, ethnic background, age and sex.

## Data Availability

Restrictions apply to the availability of these data. Data were obtained from UK Biobank and are available from the authors with the permission of UK Biobank.

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
