# Peer review of "Elevated Glycated Haemoglobin (HbA1c) Is Associated with an Increased Risk of Pancreatic Ductal Adenocarcinoma: A UK Biobank Cohort Study"

_cancers, 2023, doi:10.3390/cancers15164078_

Round 1
Reviewer 1 Report (Previous Reviewer 4)
Authors answer to my comments but the limitation sof this study are still too large , even if they were discussed and mentioned. As I do not see a chance of further improvement with data authors have, I will recommend accept the manuscript when editor in chief find it good
This manuscript is a resubmission of an earlier submission. The following is a list of the peer review reports and author responses from that submission.
Round 1
Reviewer 1 Report
Overall, the manuscript focuses on the role of HbA1c as a diagnostic biomarker for pancreatic ductal adenocarcinoma (PDAC) using the UK Biobank large cohort. The authors found that HbA1c concentrations equal to or greater than 48mmol/mol are associated with a higher risk of PDAC compared to lower HbA1c levels.
However, I still have some concerns regarding this manuscript:
Regarding the novelty and differentiation of this study, the manuscript should clearly highlight how it contributes to the existing literature, especially in light of the publications you mentioned. For example, the authors should emphasize any unique aspects of their study, such as the utilization of the UK Biobank cohort and any distinctive methodological or analytical approaches. BMI and HbA1c are metabolic markers for pancreatic cancer: Matched case-control study using a UK primary care database. https://doi.org/10.1371/journal.pone.0275369; Elevated hemoglobin A1c is associated with the presence of pancreatic cysts in a high-risk pancreatic surveillance program https://bmcgastroenterol.biomedcentral.com/articles/10.1186/s12876-020-01308-w
To strengthen the conclusions drawn from the UK Biobank cohort, it is important to validate the findings using an independent cohort or dataset. This will enhance the robustness and generalizability of the results.
If Figure 2 includes a comparison between different groups, the manuscript should provide information on the statistical significance of the observed differences. Including p-values or any relevant statistical test results, either on the figure itself or in the figure legend, will help clarify the significance of the findings.
The manuscript could discuss the potential relationship between HbA1c levels and overall survival predictions in PDAC and the expression of HbA1c in PDAC. If the study did not specifically investigate this aspect, the authors should acknowledge it as a limitation and propose future research directions to explore the association between HbA1c and overall survival outcomes in PDAC.
It would be informative to include additional figures or analyses that illustrate the risk of incident PDAC stratified by the presence or absence of diabetes mellitus (DM). This would provide insights into how HbA1c influences PDAC risk in individuals with and without DM, offering a more comprehensive understanding of the findings.
Addressing these concerns will improve the clarity, novelty, and validity of the manuscript.
None
Reviewer 2 Report
|
The introduction provides enough background and includes all relevant references. The research design is appropriate. The methods are adequately described. The results are clearly presented. The conclusions are supported by the results. |
Reviewer 3 Report
The work entitled: "Elevated glycated haemoglobin (HbA1c) is associated with an increased risk of Pancreatic Ductal Adenocarcinoma: a UK Biobank cohort study" describes the relationship between hemoglobin glycosylation and pancreatic ductal adenocarcinoma, to have predictive elements. In the emergence of early pancreatic cancer to achieve timely diagnoses. The work is interesting, well written, and the statistical analysis is adequate.
However, pancreatic cancer varies strongly among different racial groups. Although the paper corrects for body mass index by ethnicity, adenocarcinoma cases are not separated, which raises the question of whether these results are applicable in different regions. On the other hand, the article seems to disagree with the findings of a similar article titled “Association of Glycated Hemoglobin Levels With Risk of Pancreatic Cancer” (JAMA Network Open. 2020;3(6):e204945. doi:10.1001/jamanetworkopen. 2020.4945), in which ethnicity is crucial in the correlation sought. The authors should delve into these differences, considering that the strongest association is in the first 12 months of the increase in glycosylation detected.
Reviewer 4 Report
McDonnell et al investigated the relationship between HbA1c and incident Pancreas cancee using a retrospective cohort study within the UK Biobank. Participants were stratified by diabetes mellitus (DM) status, and then by HbA1c values <42mmol/mol, 42- 26 47mmol/mol, or ≥48mmol/mol. Cox proportional hazard models were used In subjects without diabeets at baseline, 12 months after recruitment, the adjusted hazard ratios for incident PDAC for HbA1c 42-47mmol/mol compared to HbA1c <42mmol/mol (reference group) was 2.14 (1.34–3.40, P=0.001); and was 8.38 (4.49–15.64, P<0.001) for HbA1c ≥48mmol/mol. The association between baseline HbA1c and incident PDAC attenuated with increasing duration of time of follow up to PDAC diagnosis.
There are two major issues:
1) HbA1c alone is not a risk factor for cancer; there is a combination of different factors, and diabetes including the fact of high HbA1c values usually need several years to impact other diseases. Authors show that after 12 months higher HbA1c is associated with higher pancreas carcinoma risk but not after 8 years. This is not logical and results from the kind of methodology. Probably, authors catch pancreas carcinoma patients with earl<y symptoms of pancreas carcinoma including increased blood sugar values. After detailed clinical investigation of these patients, physicians diagnose pancreas cancer. This is not an effect of HbA1c but vise versa; PC cause higher HbA1c as one of early symptoms. This hypothesis is even confirmed by the fact that only in first years but not after many years HbA1c is associated with cancer
2) Using <42mmol/mol, 42- 26 47mmol/mol, or ≥48mmol/mol as classes (<7, 7-7.4,>=7.5%) has no sense. When we speak about diabetes and diabetes-related direct and indirect complications, we know that no diabetes is defined as <6.5% since values <7.5 are accepted by T2D as no complications are expected as long HbA1c are not higher that 7.5%. First after 48mmol/mol, the incidence of complications as well as of everything what can be impacted by HbA1c, increased.